# Effect of Renal Denervation on the Plasma Adiponectin Concentration in Patients with Resistant Hypertension

**DOI:** 10.3390/jcm12062114

**Published:** 2023-03-08

**Authors:** Beata Czerwieńska, Michał Lelek, Damian Gojowy, Stanisław Surma, Katarzyna Mizia-Stec, Andrzej Więcek, Marcin Adamczak

**Affiliations:** 1Department of Nephrology, Transplantation and Internal Medicine, Medical University of Silesia in Katowice, Francuska Str. 20-24, 40-027 Katowice, Poland; 2First Department of Cardiology, School of Medicine in Katowice, Medical University of Silesia, 47 Ziołowa Str., 40-635 Katowice, Poland

**Keywords:** renal denervation, resistant hypertension, adiponectin

## Abstract

(1) Introduction: Adiponectin is synthetized by white adipose tissue and has anti-diabetic, anti-atherosclerotic, anti-thrombotic, anti-inflammatory, and cardioprotective properties. In patients with arterial hypertension, plasma concentration of adiponectin is lower than in healthy subjects. Renal denervation, i.e., percutaneous ablation of fibers from the sympathetic nervous system located in the wall of the renal arteries by radio frequency waves, is a method of resistant arterial hypertension treatment. (2) The aim of this single center, interventional, clinical study was to assess the effect of renal denervation on the plasma adiponectin concentration in patients with resistant arterial hypertension. (3) Materials and methods: 28 patients (13 women, 15 men) aged 54.4 ± 9.2 years with resistant hypertension who underwent renal denervation using Simplicity catheters (Medtronic, Inc., Northridge, CA, USA) were enrolled in the study. Plasma adiponectin concentration was determined using the Human Adiponectin ELISA Kit (Otsuka Pharmaceutical Co, Tokyo, Japan) before the renal denervation and 6 and 12 months after this procedure. (4) Results: Blood pressure (BP) values before renal denervation and 6 and 12 months after this procedure were as follows: systolic BP 190.4 ± 24.5, 160.8 ± 14.5, 155.7 ± 17.9 mmHg (*p* < 0.001) and diastolic BP 111.7 ± 18.9, 88.9 ± 8.3, 91.2 + 10.2 mmHg (*p* < 0.001), respectively. Body mass index (BMI) before renal denervation, 6 and 12 months after this procedure were 31.5 ± 4.2, 30.5 ± 4.4, 30.2 ± 4.0 kg/m^2^, (*p* = 0.057), respectively. Plasma adiponectin concentration before the renal denervation and 6 and 12 months after this procedure were 4.79 (3.95; 9.49), 7.58 (5.04; 9.51), 6.62 (4.57; 11.65) [µg/mL] (*p* = 0.007), respectively. (5) Conclusions: Plasma adiponectin concentration increases significantly after successful renal denervation in patients with resistant hypertension. Higher plasma adiponectin concentration may participate—beyond blood pressure reduction—in the cardiovascular benefits related to successful renal denervation; however’ clinical consequences of these results need further investigations.

## 1. Introduction

The kidneys are innervated by two types of sympathetic nervous system fibers: efferent fibers conducting impulses from the central nervous system to the kidneys and afferent fibers conducting impulses from the kidneys to the central nervous system [1]. The consequence of efferent fiber activation is sodium and water retention, vasoconstriction of renal arteries, decreased renal blood flow, and decreased glomerular filtration [2,3]. The stimulation of afferent fibers increases the activity of pressure centers in the central nervous system, which leads to an increase in peripheral resistance and blood pressure, as well as an increase in the heart rate [2]. Therefore, damage to fibers of the sympathetic nervous system during the denervation procedure may lead to a reduction in blood pressure [2,3,4,5]. One of the methods of percutaneous renal denervation is the radio frequency (RF) ablation method [3,5]. This method uses central (i.e., arterial) vascular access and the ablation procedure is performed with intravascular catheters specially designed for this purpose [3].

Adiponectin is one of the adipokines [6,7,8,9,10] synthetized by the white adipose tissue. It has a multidirectional beneficial effect [8,9], acting as an anti-diabetic, anti-atherosclerotic, anti-thrombotic, anti-inflammatory, and cardioprotective agent [8,9]. The increase in adiponectin gene expression in adipocytes is mediated by different transcription factors: PPAR (peroxisome-proliferator activated receptor) γ, C/EBP (CCAAT/enhancer-binding protein) α, SREBP (sterol-regulatory-element-binding protein)-1c, FoxO1 (forkhead box O1) and Sp1 (specificity protein 1) [11]. On the other hand, inflammatory mediators TNF-α, CRP and IL-6 lead to the decrease of adiponectin gene expression [12,13]. Moreover, it has been shown that insulin, glucocorticosteroids, hydrogen peroxide, and nicotine also inhibit adiponectin gene expression in adipocytes. [14,15]. Similarly, stimulation of sympathetic adrenergic system decreases adiponectin gene expression. It was shown that β-adrenergic agonist isoproterenol inhibited the expression of the adiponectin gene and that β-adrenergic receptors mediated this inhibitory effect [15]. In humans, plasma adiponectin concentration depends mainly on the mass of adipose tissue and on kidney function [8,9]. Moreover, women have higher plasma concentrations of adiponectin than men [16]. Higher plasma concentrations of this adipokine are also observed in subjects with normal body weight than in obese subjects [17,18]. It has been shown that weight loss, healthy diet, and physical activity restore adiponectin levels [15,19,20,21,22]. It has been also found that a number of drugs increase adiponectin concentration (temocapril, ramipril, losartan, candesartan, fenofibrate, thiazolidinediones, glimepiride, and nebivolol) [23,24,25,26,27,28,29,30,31,32,33,34,35]. In patients with essential hypertension, plasma adiponectin concentration is lower than in healthy subjects [6,36]. Results of clinical studies completed, among others, in patients with prehypertension confirm that the reduced plasma adiponectin concentration is a risk factor for the development of hypertension [37,38,39,40,41]. In addition, compared to healthy subjects, patients with arterial hypertension are characterized by a lower expression of the adiponectin fraction of high molecular weight, showing more specific anti-atherosclerotic and anti-diabetic properties than unfractionated adiponectin [42]. Low plasma concentration of this adipokine in patients with hypertension may be involved in the development of morphological changes and functional abnormalities of the heart, blood vessels, and kidneys. Thus, hypoadiponectinemia may contribute to the development of left ventricular hypertrophy and diastolic heart failure [43], advanced hypertensive changes in the retinal vessels [44], arterial stiffness [45], increased thickness of the complex intima and middle carotid artery [46], and albuminuria [47].

The results of experimental studies indicate that activation of the sympathetic nervous system inhibits the synthesis of adiponectin by white adipose tissue [48]. In clinical studies, pharmacological blockade of the sympathetic nervous system has been shown to increase plasma adiponectin concentration [49]. Moreover, hypoadiponectinemia observed in patients with type 2 diabetes is associated with high activity of the sympathetic nervous system [50].

Taking into account the above presented data, it is justified to conduct clinical studies assessing the impact of non-pharmacological intervention in reducing the activity of the sympathetic nervous system on plasma concentration of adiponectin.

The aim of the current single-center, observational, clinical study was to evaluate the effect of ablation of sympathetic nervous system fibers in the wall of renal arteries using radio frequency waves, i.e., renal denervation, on the plasma adiponectin concentration in patients with resistant hypertension.

## 2. Materials and Methods

Twenty-eight patients (13 female, 15 male) aged 54.4 ± 9.2 years with resistant primary arterial hypertension were qualified for renal denervation procedure and included in this single-center, interventional, clinical study. Secondary causes of arterial hypertension in these patients were excluded by conducting careful clinical, hormonal, and radiological examinations. Among them, the diagnostic procedures of hyperthyroidism, hypercortisolemia, pheochromocytoma, and primary hyperaldosteronism were completed. Moreover, in all patients angio-CT of renal arteries was performed in order to exclude renal artery stenosis. At baseline, patients were treated with following antihypertensive drugs: β-adrenergic receptor antagonists (all 28 patients, 100%), α-adrenergic receptor antagonists (16 of 28 patients, 57%), calcium receptor antagonists (all of 28 patients, 100%), angiotensin-converting enzyme inhibitors (20 of 28 patients, 71%), angiotensin-2 receptor antagonists (15 of 28 patients, 54%), methyldopa (2 of 28 patients, 7%), clonidine (8 of 28 patients, 29%), rilmenidine (1 of 28 patients, 4%), moxonidine (1 of 28 patients, 4%), loop diuretics (15 of 28 patients, 54%), spironolactone (11 of 28 patients, 39%), thiazide diuretics (11 of 28 patients, 39%), and thiazide-like diuretics (7 of 28 patients, 25%). Three patients (11%) were treated with four antihypertensive drugs, 7 patients (25%) with five drugs, 13 patients (46%) with six, 3 patients (11%) with seven, one patient (4%) with eight and one patient (4%) with nine antihypertensive drugs). Left ventricular hypertrophy was found in 21 patients (75%) and hypertensive retinopathy in 22 patients (79%). Eight patients (29%) had chronic kidney disease presumably as a consequence of hypertensive kidney injury with normal eGFR (>60 mL/min/1.73 m^2^) and non-nephrotic range proteinuria. Thirteen patients (46%) additionally suffered from diabetes mellitus.

Renal denervation was completed in the Interventional Cardiology Unit of 1st Department of Cardiology, Medical University of Silesia, Katowice, Poland. To prevent intraoperator variability of this procedure, all denervations were performed by a single well-trained operator (Dr Michał Lelek). A Symplicity catheter (Ardian, Palo Alto, CA, USA) was used to perform all renal denervation procedures via the femoral artery, under local anesthesia. A Symplicity catheter with a 5F diameter was inserted into the lumen of the renal artery through a vascular sheath with a 6F diameter placed in the femoral artery and a catheter with a 6F diameter placed at the outlet of the renal artery. During the procedure, the catheter was connected to a radiofrequency generator. Thus, the energy of 5–8 W was emitted into the wall of the renal artery. The aim of this procedure was to damage the fibers of the sympathetic nervous system located in the renal artery adventitia. At the beginning of the procedure, the Symplicity catheter was placed in the distal part of the renal artery trunk where the first application was made. Then, under radiological control, the catheter was withdrawn approximately by 5 mm and rotated by 90°. After such displacement and rotation of the catheter tip, another application was performed. Therefore, the applications covered the entire circumference of the renal artery. Depending on the length of the trunk within one renal artery, 4–5 applications were made. In all patients, a single procedure was performed on both sides (in left and right renal artery). In order to avoid an excessive increase in temperature in the lumen of the artery during the application, the catheter was rinsed continuously with physiological saline. Before and during the procedure, patients received heparin in a dose that allowed them to obtain the activated clotting time (ACT) in excess of 250 s. The time of the entire procedure was 45–60 min. In all studied subjects, no significant complications were observed during the periprocedural period. After the renal artery denervation procedure all patients were treated with acetylsalicylic acid (75 mg per day) for at least two weeks.

Patients were followed-up after renal denervation for one year in the Hypertension Outpatient Clinic in the Department of Nephrology, Transplantation and Internal Medicine, Medical University of Silesia Clinic Hospital in Katowice, Poland. During this time, to prevent intraobserver variability, all patients were treated by a single physician (Dr Beata Czerwieńska) and the intention was not to change pharmacological antihypertensive treatment which was recommended before renal denervation.

Two patients died during the 12-month follow-up period and were not included in further analyses. One patient (female) died because of hemorrhagic stroke. The second patient (male) died because of complications after myocardial infarction. Therefore, the final studied group consists of 26 patients (12 female, 14 male) aged 54.6 ± 8.0 years. In all patients, the following parameters were analyzed: office blood pressure, heart rate, body weight, body mass index, creatinine, fasting glucose, total cholesterol, triglycerides, CRP, adiponectin, and insulin plasma concentrations, as well as, blood concentration of glycated hemoglobin (HbA1c). The glomerular filtration rate was estimated using the short MDRD formula. Additionally, the HOMA-IR index was calculated according to the formula: fasting insulin [µU/mL] × fasting glucose [mmol/L]/22.5. In all patients, an office blood pressure measurement was performed. The above-mentioned parameters were analyzed before, 6 months, and 12 months after the renal denervation.

Blood pressure measurements were taken in the doctor’s office. The room where blood pressure measurements were taken was a quiet and peaceful place with a comfortable temperature for the patient. Blood pressure measurements were performed by a physician. There were no other people in the doctor’s office during the blood pressure measurement. For at least 30 min before the doctor’s visit and blood pressure measurement, the patient did not consume a meal, caffeine, smoke cigarettes, did not undertake physical effort or exercise, and did not use medications. Blood pressure measurements were taken after at least 5 min. of rest in a sitting position. An upper arm cuff was used to measure blood pressure, the width of which was selected individually for each patient, depending on the arm’s circumference. Blood pressure was measured using an automatic, electronic (oscillometric) blood pressure monitor. During the first examination, blood pressure was measured in both arms. Subsequent measurements were completed on the arm with higher blood pressure values. If there was no difference in pressure between the arms, the following measurements were made on the arm of the side of non-dominant hand. During the measurement, the patient remained sitting, with his back resting, the elbow of the examined limb was supported, and the arm with the blood pressure monitor’s cuff was at heart level. The lower limbs were not uncrossed, and the patient’s feet were placed freely, flat on the ground. During blood pressure measurements, the patient did not talk or move. Blood pressure was measured three times during the medical visit with a 1-min. interval between measurements. The average of the last two measurements was taken into account. This method was in line with actual practice guidelines of the European Society of Hypertension [51]. Plasma adiponectin concentration was measured with the use of human adiponectin ELISA Kit, Otsuka Pharmaceutical Co., Tokyo, Japan. Insulinemia was measured with the use of the immunoassay method automatically with Cobas analyzer, Roche, Basel, Switzerland. Plasma CRP concentration was measured with the use of an ELISA Kit from Immundiagnostik AG, Bensheim, Germany. Other parameters were measured using routine laboratory methods.

Statistical analysis was conducted using STATISTICA software. Shapiro–Wilk test was used to assess the normality of distribution. The following tests were used in statistical analysis: chi-square, Friedman’s ANOVA, and multivariable analysis. Results were presented as a mean with standard deviation (±SD), when data had parametric distribution and as a median with interquartile range (Q1;Q3).

The Medical University Ethics Committee consent number is KNW/0022/KB1/22/II/12.

## 3. Results

Office blood pressure (BP) decreased significantly after the renal artery denervation procedure: systolic BP (190.4 ± 24.5, 160.8 ± 14.5, 155.7 ± 17.9 mmHg before the procedure, 6 months after, and 12 months after, respectively; *p* < 0.001 and diastolic BP: 111.7 ± 18.9, 88.9 ± 8.3, 91.2 + 10.2 mmHg, before the procedure, 6 months after, and 12 months after, respectively; *p* < 0.001). Heart rate in mentioned intervals decreased (79.1 ± 16.8, 73.9 ± 11.3, 73.1 ± 11.4 per minute, respectively; *p* = 0.006). The number of antihypertensive drugs was the same before (6.0 (5.0; 6.0)) as 6 and 12 months after renal denervation (5.5 (5.0; 7.0) and 6.0 (5.0; 7.0), respectively).

Body weight and body mass index (BMI) decreased after the renal artery denervation procedure (88.5 ± 13.9, 87.3 ± 13.5, 86.5 ± 12.7 kg before the procedure, 6 months after, and 12 months after, respectively; *p* = 0.04 and 31.5 ± 4.2, 30.5 ± 4.4, 30.2 ± 4.0 kg/m^2^ before the procedure, 6 months after, and 12 months after, respectively; *p* = 0.057), (Table 1).

It has been shown that renal artery denervation was followed by a significant increase in plasma adiponectin concentration (4.79 (3.95; 9.49), 7.58 (5.04; 9.51), 6.62 (4.57; 11.65) [µg/mL]; *p* = 0.007); before the procedure, 6 months after, and 12 months after procedure, respectively) (Table 1, Figure 1). Plasma insulin concentration decreased significantly after the procedure (20.8 (16.3; 39.2), 16.0 (11.9; 20.7), 14.4 (12.6; 19.4) [µIU/mL]; *p* = 0.001, respectively), but there was no significant difference in the HOMA index during the follow-up period (Table 1).

In the studied group there was no significant correlation between BMI as well as systolic/diastolic blood pressure and plasma adiponectin concentration before denervation as well as 6 and 12 months. There was also no correlation in concentration of plasma adiponectin and glucose 6 and 12 months after denervation.

Two multivariable analyses were carried out. In the first, plasma adiponectin concentration after 12 months was a dependent value. Baseline plasma adiponectin concentration and glucose, body mass index, eGFR after 12 months as an independent value. It has been shown that plasma adiponectin concentration after 12 months depends only on plasma adiponectin concentration at baseline. In the second analysis, plasma adiponectin concentration after 12 months was a dependent value. Baseline plasma adiponectin concentration and change in glucose plasma concentration, body mass index, eGFR in 12 months as an independent value. It has been shown that plasma adiponectin concentration after 12 months depends also only on plasma adiponectin concentration at baseline.

Change of plasma adiponectin concentration at 6 and 12 months in multivariable analysis did not depend on change of BMI (*p* = 0.9 and *p* = 0.1), systolic blood pressure (*p* = 0.6 and *p* = 0.5), glucose plasma concentration (*p* = 0.8 and *p* = 0.7) and eGFR (*p* = 0.7 and *p* = 0.09) at 6 and 12 months respectively.

Only one patient was treated with moxonidine. Baseline blood pressure in this patient was 190/93 mmHg, 6 months after denervation 153/76 mmHg and 12 months after 147/84 mmHg. Therefore, there were decrease of systolic and diastolic blood pressure in 6 months 37/17 mmHg, and after 12 months 34/9 mmHg, respectively. One patient treated with rilmenidine had blood pressure 170/110 mmHg at baseline, 157/82 mmHg after 6 months and 138/83 mmHg after 12 months. Therefore, decrease of systolic and diastolic blood pressure in 6 months were 13/28 mmHg, and after 12 months 32/27 mmHg, respectively (mean decrease in other patients 31/25 mmHg and 35/20 mmHg, respectively). Because of the low number of patients treated with rilmenidine and moxonidine (only two patients) formal comparison between patients treated with rilmenidine or moxonidine and the other patients is not possible. However, it seems that the antihypertensive effect of rilmenidine and moxonidine is similar to other drugs.

Patients were divided into two groups: responders, with a systolic BP after 6 months from the procedure lower at least 25 mmHg in comparison to systolic BP before the procedure, and low-responders whose systolic BP change after denervation was lower than 25 mmHg. Low-responders were characterized by lower baseline systolic BP (174.7 vs. 203.9 mmHg) and lower GFR (77.2 vs. 95.2 mL/min/1.73 m^2^) as well as higher body weight (83.7 vs. 93.6 kg) and BMI (34.3 vs. 29.3 kg/m^2^), in comparison to responders. Plasma adiponectin concentration increased significantly in responders during the follow-up period (6.23 (3.83; 9.79), 8.18 (5.04; 10.64), 8.88 (4.57; 14.51) [µg/mL]; *p* = 0.03, before denervation, 6 months after and 12 months after, respectively) (Figure 2) and did not change significantly in low-responders (4.65 (4.22; 6.86), 6.42 (4.93; 8.84), 5.98 (4.46; 9.05) [µg/mL]; *p* = 0.2) (Figure 3). There was also a more pronounced decrease of plasma insulin in patients who were responders (19.6 (12.9; 45.6), 13.7 (11.3; 16.1), 13.8 (9.6; 15.2) [µIU/mL], *p* = 0.03) in comparison with low-responders (23.5 (17.0; 37.4), 18.8 (16.1; 26.1), 18.9 (13.4; 24.4) [µIU/mL], *p* = 0.01). HOMA-IR index did not change significantly in both subgroups (Table 2).

## 4. Discussion

In the present study, we were able to document the significant increase in plasma adiponectin concentration in patients with resistant hypertension after renal denervation.

The relationship between renal artery denervation and plasma adiponectin concentrations has so far been investigated in only a few clinical studies. In a study by Eikelis et al. of 57 patients with resistant hypertension who underwent renal denervation procedure, it was found that three months after the procedure plasma adiponectin concentration was higher than before renal denervation [52]. On the other hand, different results were obtained in the study by Miroslawska et al., who observed 20 patients with resistant arterial hypertension. There was no significant effect of renal denervation on plasma adiponectin concentrations in patients with refractory hypertension after two years of follow-up [53]. Based on the blood pressure data given in this study, it seems that the lack of effect of renal artery denervation may be due to a non-effective procedure.

Heterogeneity of the above-mentioned results may be due heterogeneity of the completion of the ablation procedures. In the current study, systolic blood pressure decrease in 12 months follow-up was from 190.4 ± 24.5 to 155.7 ± 17.9 mmHg, in Mirosławska et al. study 164 ± 21 to 150 ± 18 mmHg in 2 years and in Eikelis from 168.8 ± 2.6 to 155.2 ± 3.2 mmHg in 3 months. One of the reasons for the heterogeneous antihypertensive effects of the procedure may be possible differences among the subjects in the anatomy of sympathetic nervous fibrous in renal artery adventitia. Moreover, it should be stressed that during the procedure is not possible to control directly the completeness of denervation in vivo [54].

Adiponectin participates in the regulation of blood pressure through two mechanisms: by influencing the activity of the sympathetic nervous system [55] and by stimulating the release of nitric oxide within the vascular endothelium [56]. In animal studies it was shown that administration of adiponectin both to the cells of the central nervous system and by the intravenous route inhibits the activity of the renal fibers of the sympathetic nervous system and reduces blood pressure [55]. Such action is the result of the stimulation of adiponectin receptors located in the hypothalamic supraspinate nucleus and the nucleus of the lonely spinal cord. In animals, related to adiponectin blood pressure lowering effect can be blocked by damaging the suprachiasmatic nucleus. The results of experimental studies showed that sympathetic nervous system overstimulation may result in reduction of adiponectin release from adipocytes [57]. The increase in plasma adiponectin concentration shown in our study is most likely due to effective renal denervation, because we found that the change in the concentration of this adipokine occurred only in patients with more pronounced decrease of blood pressure, i.e., with presumably more effective renal denervation procedure. There is also other evidence in the literature on the relationship between sympathetic nervous system activity and adiponectin release. Nowak et al. treated 20 patients with essential hypertension with rilmenidine (selectively acts on I_1_-imidazoline receptors in the medulla vasomotor center and the kidneys and leads to reduction of sympathetic nerve system activity) at a dose of 1–2 mg/day. After six months of such treatment, there was a significant increase in plasma adiponectin concentration without significant changes in insulin sensitivity and body fat content [49].

Increasing the adiponectin concentration in patients with arterial hypertension may be of clinical importance. Adiponectin is an adipose tissue hormone with multidirectional beneficial effects on the circulatory system and metabolism [58]. A meta-analysis of 12 prospective studies by Zhang et al. showed that higher adiponectin concentrations were associated with a lower risk of coronary heart disease (RR = 0.83; 95% CI: 0.69–0.98) [59]. Moreover, a meta-analysis of 13 prospective studies by Li et al. found that the relative risk of type two diabetes was 0.72 (95% CI: 0.67–0.78) per 1—log μg/mL increment in adiponectin concentrations [60]. The anti-diabetic properties of adiponectin were confirmed in a meta-analysis of 34 prospective studies by Wang et al. This meta-analysis found that the relative risk for diabetes type two was 0.53 (95% CI: 0.47–0.61) in subjects with the highest adiponectin tercyl [61]. Moreover, as demonstrated by Chen et al. in a study of 101 hypertensive patients, hypoadiponectinemia was related to peripheral arterial stiffness [62].

In the current study, patients who showed significant office blood pressure reduction (responders) after renal denervation were characterized by increased adiponectin concentration during the 12-months follow-up observation period besides no significant change in BMI (Table 2). On the other hand, results of multivariate analysis did not confirm the relationship between plasma adiponectin and SBP. However, because of the low number of subjects in studied group, lack of this relation is not proven but only supposed. It seems that increase in plasma adiponectin concentration in responders was not related to changes of BMI, because increase in plasma adiponectin concentration in this group was not accompanied by significant BMI changes. The decrease in BMI in low-responders was statistically significant but did not translate into an increase in adiponectin concentration (Table 2, Figure 3). It should also be noted that doses and numbers of diuretic agents were not changed during the follow-up time, so BMI seems not to be affected by the change of volemia in both groups (responders and low-responders).

The major limitation of the current study is a lack of a placebo group and a low number of studied patients. On the other hand, because of its single-center study character, the procedure of denervation was performed by the same operator in one medical center, which allowed to avoid, as much as possible, bias related to the procedure. Further observation of these patients completed by single physician helped to standardize the approach to antihypertensive treatment and non-pharmacological recommendations in long-term follow-up.

Results of the current study may open a discussion about the impact of adiponectin concentration and its reduction after renal denervation on the risk of cardiovascular complications in patients with resistant arterial hypertension. Additional, long-term studies are needed.

## 5. Conclusions

Plasma adiponectin concentration increases significantly after successful renal denervation in patients with resistant hypertension. Higher plasma adiponectin concentration may participate—beyond blood pressure reduction—in the cardiovascular benefits related to successful renal denervation, however clinical consequences of these results need further investigations.

## Figures and Tables

**Figure 1 jcm-12-02114-f001:**
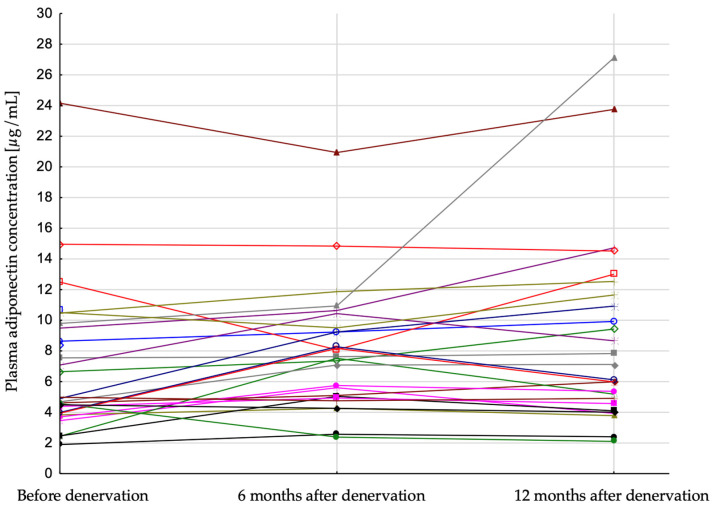
Plasma adiponectin concentration in patients after renal artery denervation in all studied patients. Each line represent result of single patient.

**Figure 2 jcm-12-02114-f002:**
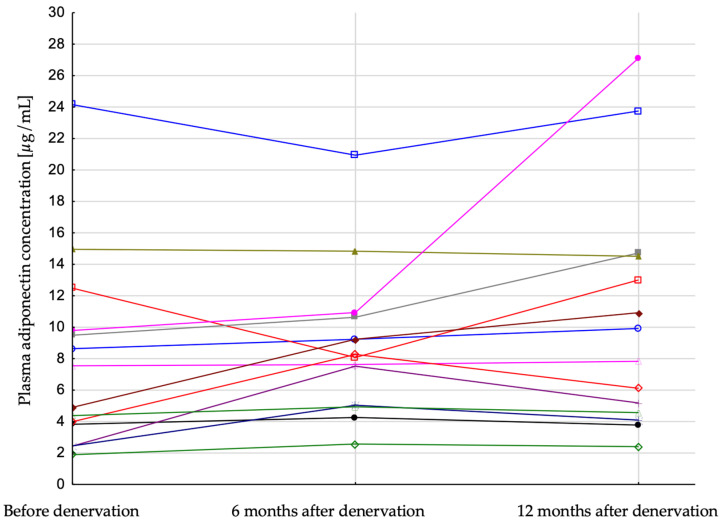
Plasma adiponectin concentration in patients with resistant hypertension before, 6 and 12 months after renal artery denervation in responders. Each line represent result of single patient.

**Figure 3 jcm-12-02114-f003:**
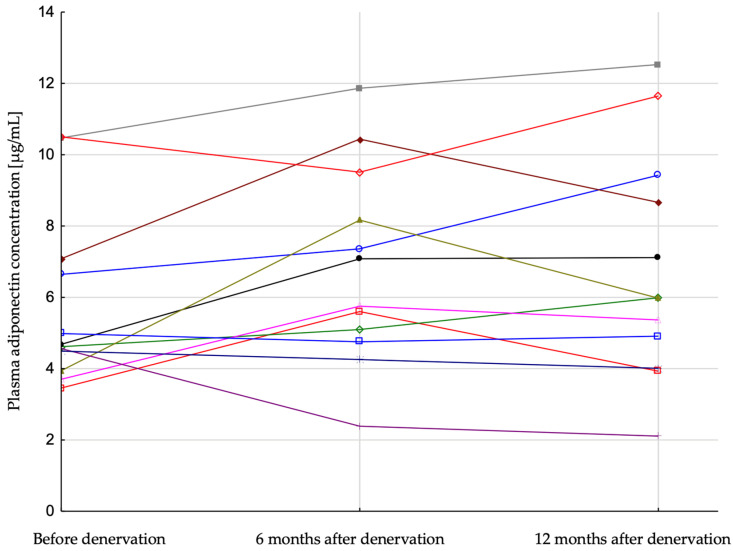
Plasma adiponectin concentration in patients with resistant hypertension before, 6 and 12 months after renal artery denervation in low-responders. Each line represent result of single patient.

**Table 1 jcm-12-02114-t001:** Characteristic of patients before, 6 months and 12 months after renal artery denervation.

Parameter	before Denervation	6 Months after Denervation	12 Months after Denervation	Statistical Significance
SBP [mmHg]	190.4 ± 24.5	160.8 ± 14.5	155.7 ± 17.9	*p* < 0.001
DBP [mmHg]	111.7 ± 18.9	88.9 ± 8.3	91.2 ± 10.2	*p* < 0.001
HR [n/min]	79.1 ± 16.8	73.9 ± 11.3	73.1 + 11.4	*p* = 0.006
Body weight [kg]	88.5 ± 13.9	87.3 ± 13.5	86.5 ± 12.7	*p* = 0.04
BMI [kg/m^2^]	31.5 ± 4.2	30.5 ± 4.4	30.2 ± 4.0	*p* = 0.057
eGFR [mL/min/1.73 m^2^]	87.5 ± 18.1	95.3 ± 18.1	93.4 ± 14.9	*p* = 0.22
Glucose [mmol/L]	5.10 (4.40; 6.30)	5.30 (4.81; 6.58)	5.66 (5.17; 7.64)	*p* = 0.20
Total cholesterol [mmol/L]	5.18 ± 1.32	4.91 ± 1.47	5.23 ± 1.12	*p* = 0.67
Triglycerides [mmol/L]	1.97 ± 0.94	1.49 ± 0.80	1.72 ± 0.92	*p* = 0.14
Glycated hemoglobin [%]	5.46 (5.08; 5.91)	5.62 (5.23; 6.05)	5.88 (5.46; 6.28)	*p* = 0.23
CRP [ng/mL]	1.5 (0.9; 4.4)	3.2 (1.5; 11.8)	2.2 (1.2; 6.4)	*p* = 0.42
Insulin [µIU/mL]	20.8 (16.3; 39.2)	16.0 (11.9; 20.7)	14.4 (12.6; 19.4)	*p* = 0.001
HOMA index	6.8 ± 5.4	5.0 ± 2.7	4.8 ± 2.4	*p* = 0.17
Adiponectin [µg/mL]	4.79 (3.95; 9.49)	7.58 (5.04; 9.51)	6.62 (4.57; 11.65)	*p* = 0.007

Grey background in cell—significant trend in time intervals (*p* < 0.1); SBP—systolic blood pressure; DBP—diastolic blood pressure; HR—heart rate; BMI—body mass index; eGFR—estimated glomerular filtration rate; CRP—C-reactive protein; HOMA—homeostatic model assessment of insulin resistance.

**Table 2 jcm-12-02114-t002:** Characteristic of responders and low-responders.

Parameter	Responders (*n* = 14)			Low-Responders (*n* = 12)		
Age [years]	52.7 ± 9.2			56.8 ± 6.1		
Male/female	7/7			7/5		
Time from denervation	Before	6 months after	12 months after	Before	6 months after	12 months after
SBP [mmHg]	203.9 ± 21.5 *	159.9 ± 13.0	159.0 ± 15.6	174.7 ± 17.7 *	161.8 ± 16.6	151.8 ± 20.3
DBP [mmHg]	115.6 ± 22.7	88.1 ± 7.1	94.0 ± 9.6	107.1 ± 12.7	89.9 ± 9.8	87.9 ± 10.4
HR [n/min]	77.4 ± 14.5	71.8 ± 9.6	72.7 ± 10.9	81.2 ± 19.5	76.4 ± 13.0	73.5 ± 12.5
Body weight [kg]	83.7 ± 12.8	82.6 ± 12.4	82.8 ± 12.2	93.6 ± 13.5	92.8 ± 13.2	90.8 ± 12.3
BMI [kg/m^2^]	29.3 ± 3.7 *	28.6 ± 3.9 *	28.8 ± 4.2 *	34.3 ± 3.0 *	32.7 ± 4.1 *	31.9 ± 3.2 *
eGFR [mL/min/1.73 m^2^]	95.2 ± 18.2 *	99.3 ± 12.9	90.1 ± 15.2	77.2 ± 12.7 *	89.1 ± 24.0	99.4 ± 13.9
Glucose [mmol/L]	4.78 (4.4; 5.1) *	4.81 (4.68; 5.47) *	5.18 (4.82; 5.98) *	5.45 (5.13; 6.79) *	6.30 (5.10; 7.12) *	6.49 (5.66; 7.94) *
Total cholesterol [mmol/L]	5.34 ± 1.31	5.70 ± 1.46	5.41 ± 1.18	5.00 ± 1.37	4.21 ± 1.13	5.01 ± 1.07
Triglycerides [mmol/L]	1.46 ± 0.76 *	1.80 ± 1.05	1.75 ± 1.11	2.47 ± 0.85 *	1.22 ± 0.40	1.69 ± 0.68
Glycated hemoglobin [%]	5.17 (4.96; 5.78) *	5.37 (5.16; 5.53)	5.54 (5.33; 6.02)	5.83 (5.64; 7.11) *	5.87 (5.71; 7.33)	6.14 (5.70; 7.56)
CRP [ng/mL]	1.75 (0.9; 4.4)	2.1 (1.7; 6.8)	2.35 (1.7; 4.0)	1.5 (0.8; 4.5)	4.45 (1.2; 13.6)	1.75 (1.0; 11.0)
Insulin [µIU/mL]	19.6 (12.9;45.6)	13.7 (11.3; 16.1) *	13.8 (9.6; 15.2) *	23.5 (17.0; 37.4)	18.8 (16.1; 26.1) *	18.9 (13.4; 24.4) *
HOMA index	5.85 ± 4.71	4.00 ± 3.07 *	3.62 ± 2.09 *	8.04 ± 6.12	6.32 ± 1.55 *	6.36 ± 1.86 *
Adiponectin [µg/mL]	6.23 (3.83; 9.79)	8.18 (5.04; 10.64)	8.88 (4.57; 14.51)	4.65 (4.22; 6.86)	6.42 (4.93; 8.84)	5.98 (4.46; 9.05)

* *p* < 0.05 between responders and low-responders; Grey background in cell—significant trend in time intervals (*p* < 0.1); SBP—systolic blood pressure; DBP—diastolic blood pressure; HR—heart rate; BMI—body mass index; eGFR—estimated glomerular filtration rate; CRP—C-reactive protein; HOMA—homeostatic model assessment of insulin resistance.

## Data Availability

The data presented in this study are available on request from the corresponding author. The data are not publicly available due to privacy.

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
