# Peer review of "Effect of Renal Denervation on the Plasma Adiponectin Concentration in Patients with Resistant Hypertension"

_jcm, 2023, doi:10.3390/jcm12062114_

Round 1

Reviewer 1 Report

This is a very interesting study aimed to assess the effect of renal denervation on the plasma adiponectin concentration in patients with resistant arterial hypertension. Below I would like to highlight some points of the study:

- Page 2, second paragraph: Could you please present the factors affecting adiponectin levels?

- Page 3, line 149: Could you please describe in detail the blood pressure measurement?

- Could you please provide data if the response after renal denervation was greater in subjects receiving drugs affecting sympathetic tone, such as rilmenidine and moxonidine?

- In tables 1 and 2 all variables are presented as mean ± standard deviation, this means that they all have a normal distribution? Αdiponectin values, if not normally distributed, it could probably be presented as median (Q1,Q3).

-Page 4, table 1 , column five. Could you please provide “p value” where there is “ NS”.

-Page 4. Could you please provide statistical analysis after adjustment for confounders affecting adiponectin levels?

- In figure 2 (page 6) it can be seen that the 4th line from the top (in purple) has an adiponectin value just under 10μg/mL which triples in 12 months after renal denervation. What are the characteristics of this person? Does its own value as an outlier raise the adiponectin mean value and affect statistical significance? In this figure, it is visually shown that 6 out of 14 people increase the value of adiponectin (percentage less than 50% and one of them that triples the value is the 4th from top to bottom….). If that person is removed, do the results change? If so, there is a risk of bias.

- Table 2 shows that Low-responders had higher Glycated hemoglobin compared to Responders, which increases more at 12 months. Does the presence of diabetes mellitus and the increase in glucose in this group affect the level of adiponectin? Could you please specify?

- In table 2, could you please provide the mean age and the male to female ratio of the responders and low-responders, respectively?

- On page 8, lines: 333-337: "The increase in plasma adiponectin concentration.... renal denervation procedure.” This conclusion is not proven sufficiently by the statistical analysis presented in the paper. Adiponectin levels should be adjusted for possible factors affecting its values (blood pressure, BMI, glucose etc) that changed from the time of renal denervation to 12 months.

-Page 9, lines 358-361: “The decrease in BMI in low-responders …..to a change in the body weight.” This is a speculation, mostly because of the small sample size. I think it should be rephrased or removed. Ideally, adiponectin levels should be adjusted by BMI.

- Page 9, lines 372-375:  “Based on results of the current study …..in patients with resistant arterial hypertension”. The reduction of cardiovascular complications in patients with resistant arterial hypertension is not proven by this small study and it is quite overestimated to assume such a thing. Furthermore, patients should have long-term close follow-up to draw such a conclusion. Could you please modify the conclusion?

- Finally, it would be interesting to compare 2 groups: the one group with increase in adiponectin levels at 12 months and the second group without increase. Was such an analysis done? In what factors do these 2 groups differed?

Author Response

Responses to the reviewer in the attachment

Reviewer 2 Report

The authors have produced a well written paper regarding the evolution of the plasma adiponectine concentration in patient treated with an ( unilateral ) RDN procedure for therapy resistant hypertension. The clinical relevance however remains to be established however.

A few remarks and suggestions

1.      In the patient group studied there is no patient being treated with Aldactone ( Spironolactone ) an EBM – guided essential drug in the treatment of therapy resistant hypertension. Is this correct ? Were 24 h RR measurements collected ? Do we have an idea about dipping and D/N variability ?

2.      Line 105 should be eGFR < 60 mL …

3.      Line 106-107 – 108 typo … should be deleted

4.      Line 126 : is the statement correct that RDN procedure was-in all but one patient- performed unilaterally ? Was the RDN procedure performed with the first gen RDN catheter ?  If so why ? Much has been written concerning the lack of effect in a unilateral RDN ( due to crosstalk mechanisms ). If results are being analyzed purely in a patient cohort with an unilateral RDN one cannot state that the procedure was performed according to the current guidelines or even state that is was effective. How do the authors explain this different outcome compared with the existing literature ?

5.      Line 210 and beyond : identification of good responders and suboptimal responders : is this correct if we are talking about an unilateral RDN ? This definition – if RDN was performed bilateral ) – is not really an accepted one. Authors should clearly state this. Possibly an emphasis on the non dichotomous nature of the rdn effect is wise.

6.      Line 356 patient who showed a significant office BP reduction after an unilateral RDN procedure …

7.      Line 365 typo “is a lack of …”

8.      Do the authors have any comparison with RDN being performed with the Spyral catheter aiming to have more expensive RDN result ?

Best regards

Author Response

(The authors gave the same response as above.)

Reviewer 3 Report

This single-center, observational clinical study evaluated the effect of renal denervation on the plasma adiponectin concentration in patients with resistant hypertension. Line 141 mentioned two patients died during the follow-up period, please add the causes of death.

Author Response

(The authors gave the same response as above.)

Round 2

Reviewer 1 Report

I thank the authors for the revised manuscript. I suppose that in lines 260, 261 and 262 where there is "12" the correct number is "6". Please correct it in the final manuscript.

Reviewer 2 Report

the authors did appropriate changes